# The association between HIV diagnosis disclosure and adherence to anti-retroviral therapy among adolescents living with HIV in Sub-Saharan Africa: A systematic review and meta-analysis

**Melkamu Merid Mengesha**[1]*, **Awugchew Teshome**[1], **Dessalegn Ajema**[1], **Abera Kenay Tura**[2], **Inger Kristensson Hallström**[3], **Degu Jerene**[3,4]

**1** School of Public Health, College of Medicine and Health Sciences, Arba Minch University, Arba Minch, Ethiopia, **2** School of Nursing and Midwifery, College of Health and Medical Sciences, Haramaya University, Harar, Ethiopia, **3** Faculty of Medicine, Department of Health Sciences, Child and Family Health, Lund University, Lund, Sweden, **4** KNCV Tuberculosis Foundation, Hague, The Netherlands

* melkamumrd@gmail.com

## Abstract

### Introduction

Nine in ten of the world's 1.74 million adolescents living with human immunodeficiency virus (ALHIV) live in Sub-Saharan Africa. Suboptimal adherence to antiretroviral therapy (ART) and poor viral suppression are important problems among adolescents. To guide intervention efforts in this regard, this review presented pooled estimates on the prevalence of adherence and how it is affected by disclosure of HIV status among ALHIV in Sub-Saharan Africa.

### Methods

A comprehensive search in major databases (Excerpta Medica database (EMBASE), PubMed, Ovid/MEDLINE, HINARI, and Google Scholar) with additional hand searches for grey literature was conducted to locate observational epidemiologic studies published in English up to November 12, 2022 with the following inclusion criteria: primary studies that reported disclosure of HIV status as an exposure variable, had positive adherence to ART as an outcome, and conducted among adolescents and children. The COVIDENCE software was used for a title/abstract screening, full-text screening, the JBI quality assessment checklist, and data extraction. Random effects model was used to pool estimates. Furthermore, sensitivity analysis and subgroup analysis were also conducted by age groups and type of adherence measures used.

### Results

This meta-analysis combines the effect estimates from 12 primary studies with 4422 participants. The prevalence of good adherence to ART was 73% (95% CI (confidence interval):

**Data Availability Statement:** All relevant data are within the paper and its Supporting Information files.

**Funding:** The author(s) received no specific funding for this work.

**Competing interests:** The authors have declared that no competing interests exist.

56 to 87; $I^2$ = 98.63%, P = <0.001), and it was higher among adolescents who were aware of their HIV status, 77% (95% CI: 56 to 92; $I^2$ = 98.34%, P = <0.001). Overall, knowledge of HIV status was associated with increased odds of adherence (odds ratio (OR) = 1.88, 95% CI: 1.21 to 2.94; $I^2$ = 79.8%, P = <0.001). This was further supported in a subgroup analysis by age (seven studies, pooled OR = 1.89, 95% CI: 1.06 to 3.37; $I^2$ = 81.3%, P = <0.0001) and whether primary studies controlled for confounding factors (six studies provided adjusted estimates, pooled OR = 2.61, 95% CI: 1.22 to 5.57; $I^2$ = 88.1%, P = <0.001) confirmed this further.

## Conclusions

Our meta-analysis and systematic review revealed that knowledge of one's HIV status was associated with adherence to ART, particularly among adolescents. The findings underscored the importance of encouraging disclosure in order to enhance adherence among adolescents.

## Introduction

Globally, adolescents represent a growing share of people living with HIV. In 2019, 1.74 million adolescents were living with HIV, with the Sub-Saharan Africa accounting 1.5 million (88%). While there have been substantial declines in new infections amongst adolescents, adolescents still account for about 5% of all people living with HIV and about 10% of new adult HIV infections [1]. In 2019 alone, 170,000 adolescents were newly infected with HIV and 34,000 adolescents died of AIDS-related causes [2].

The increasing availability and effectiveness of ART worldwide has enabled many adolescents to reach adulthood and achieve their goals [3,4]. However, many challenges with adolescent HIV treatment remain, and often place this age group at risk having poorer outcomes than children and adults across the HIV care cascade, including lower adherence, poorer retention in care, lower rates of virological suppression and higher rates of mortality [5–9]. Globally, from adolescents living with HIV (ALHIV), only 43% are engaged in care, 31% are retained in care and a dismal 30% are virally suppressed [10].

For ART to work effectively with successful outcomes, a high level of adherence must be achieved, varying between 80% and 95% depending on the specific medications being used [11,12]. However, ART adherence and retention in care are generally suboptimal in low resource settings [13–17]. Worldwide, only 62% of adolescents on ART therapy achieve an adherence rate of at least 85% [18,19] and non-adherence remains the single most significant challenge in ALHIV [1].

Currently, disclosure of HIV diagnosis to infected children and adolescents has become important issue in clinical practice, as it presents with many clinical and psychosocial benefits that seek to improve the quality of HIV care [20]. In the early dates of the epidemic, especially in the Sub-Saharan Africa where access to ART was limited, and survival rate was very low, few providers were concerned about disclosing the diagnosis result to these children and adolescents [21]. However, in recent years, improved access to ART and its success as lifelong HIV-care has necessitated a shift in non-disclosure practices, and parents/caregivers and health care providers are expected to disclose to infected children and adolescents [22]. Contrary to this expectation, the increased survival times has rather presented with one of the

biggest psychosocial challenges that caregivers face with regards to the disclosure of HIV diagnosis to their infected children and adolescents [22–25]. As such more caregivers are hesitant or unable to disclose and many might choose to withhold an HIV diagnosis throughout the HIV-infected child's life [20]. Parents and caregivers struggle with disclosing HIV to their children, typically deferring until adolescence to explain how and why they have HIV due to concerns about their reaction and worry that they would tell others about the family's HIV status [26].

Several research studies in resource-limited settings have assessed the association between disclosure status and levels of ART adherence. However, there was limited conclusive evidence regarding the effects of disclosure on treatment adherence as the previous research evidences reported both advantages and disadvantages associated with HIV status disclosure. For instance, Grimsrud et al. reported that disclosure was associated with an increase in medication adherence, mediated through increased social support [27]. Contrary to this, a systematic review by Hudelson and Cluver [28] reported that non-disclosure of diagnostic result is associated with good adherence, while another review by Nichols et al. found no conclusive evidence on the impact of disclosure on adherence [29]. Spurred on by the UNAIDS' Fast Track 90–90–90 targets which aimed to have 90% of all ALHIV diagnosed, 90% of those diagnosed HIV-positive linked to HIV care continuum, and 90% of those adolescents receiving treatment achieving viral suppression by 2020 [30], interventions to improve adherence are required to achieve targeted viral suppression. Hence, this systematic review and meta-analysis aimed to pool estimates on the level of adherence and how it is affected by disclosure of HIV status.

## Methods

### Protocol and registration

This systematic review and meta-analysis was conducted following guidelines of the Preferred Reporting Items for Systematic Reviews and Meta-Analysis (PRISMA) [31]. It is registered in the International Prospective Register of Systematic Reviews (PROSPERO) which is available at (https://www.crd.york.ac.uk/prospero/display_record.php?ID=CRD42020178084). We published the protocol, and it can be accessed online at https://doi.org/10.1186/s13643-020-01420-8 [32]. See (S1 Table) for the completed PRISMA 2020 checklist.

### Eligibility criteria

Study design/characteristics: Observational studies (analytic cross-sectional, case-control, and cohort) that reported association between disclosure and adherence to ART among ALHIV and reported odds ratio (OR) as a measure of association or allowed computing it from the data were considered for inclusion. In this review, we considered studies conducted in Sub-Saharan African countries and published in English; Last updated check was done on November 12, 2022.

Population: The primary studies considered for pooling measured disclosure and adherence information of children and adolescents (ages between 10 to 19 years) who acquired HIV either through perinatal infection or behaviorally. Source of information could be either the primary caregiver or adolescents themselves. Primary caregivers are adults over the age of 18 years who lived in the same household as the adolescent and able to report about the HIV/AIDS-related care that the adolescent received.

Exposure: The primary exposure of interest was disclosure of HIV status (children and adolescents know their HIV status). Studies reporting adolescents own HIV status disclosure by caregivers and the adolescent's onward disclosure of his/her HIV status to significant others including friends and families were eligible for inclusion. However, full disclosure which is

limited in the HIV/AIDS care in Su-Saharan Africa [33] is defined when the adolescent knows the name of his/her illness (HIV and/or AIDS), received disease-specific information (for example, how the virus is transmitted and ways of prevention), and knows how they acquired the infection [34].

Comparators (controls): included studies compared the adherence in the exposed group (adolescents who knew their HIV status or were reported to know) against that in the unexposed group (adolescents who did not knew of their HIV status or were reported that they did not know).

Outcome: adherence to ART based on self-report (caregiver or adolescent) or based on pill-count (home-based unannounced or clinic based) was considered as study outcome. We also considered prevalence reports of HIV status disclosure and reports of level of adherence to ART. Despite its limitation of overestimating adherence, self-reported measures of adherence still have clinical value in predicting viral load and hence screening for poor adherence [35,36]. Consequently, in this systematic review and meta-analysis, we considered all primary studies that reported adherence based on self-report, pill-count, or mixed methods of adherence measures. Good or optimal adherence was based percent of medication doses taken (≥95%) and, in this meta-analysis, adherence measure reports of the last 30 days or 7 days was considered for pooling.

## Data source and search strategy

A comprehensive search was conducted in major databases – PubMed/MEDLINE, Excerpta Medica database (EMBASE), Ovid/MEDLINE – using key terms and Medical Subject Headings (MeSH) specifically designed for the respective databases. To access subscription-based articles, the World Health Organization HINARI database was also searched. Furthermore, to access grey literature, hand search of institutional repositories and systematic search on Google Scholar was conducted. The key search terms used to build the search strings included "adolescents", "disclosure", "self-disclosure", "adherence", "antiretroviral therapy", and "Sub-Saharan Africa". Besides the systematic searches employed, a bibliographic search of identified articles was conducted to identify additional articles. The complete list of the search strategy and key terms were summarized in an additional file (S2 Table).

## Data management and study selection

Articles retrieved from databases were exported to EndNote version 9.1 citation manager and then to COVIDENCE, a software for a systematic review production tool for title/abstract screening, full-text screening, data abstraction, and quality assessment [37]. The COVIDENCE software removes duplicates and the title and abstract screening and full-text reviews were conducted independently by two reviewers [MM and AT /or AKT or DA]. Any disagreement between two reviewers was resolved by a third reviewer assigned to resolve conflicts [MM or AKT] and then final decision was made either to include or exclude an article.

## Data extraction and quality assessment

Data extraction and quality assessment was also conduct using COVIDENCE [37]. Two authors, MMM and AT, independently extracted data on study identification, methods, population, exposure, and outcome. The Joanna Brigs Institute's (JBI) critical appraisal tools for analytic cross-sectional study and case-control study were used to assess the quality of studies [38]. This tool has eight items for the assessment of analytic cross-sectional studies and ten items for that of case-control studies [38]. The risk of bias interpretation was made as following: analytic cross-section study (If an article scored 7–8 points, low risk of bias; 4–6 points,

moderate risk of bias; 0–3 points, high risk of bias); case control (9–11: Low risk of bias; 5–8: Moderate risk of bias; 0–4: High risk of bias); cohort(9–11: Low risk of bias; 5–8, moderate risk of bias; 0–4, high risk of bias). See (S3 Table) for results of study quality assessment based on the Joanna Briggs Institute's critical appraisal checklists for observational studies.

## Data synthesis and statistical analysis

Summary of pooled estimates were presented graphically in terms of forest plots and the visual assessment of publication bias. Study characteristics of the included primary studies were summarized and presented in a table. The summary study characteristics table briefly presents the following information: author name, publication year, country, study setting and design, population studied, adherence measures used, adjustments for confounding, risk of bias, and major findings. STATA version 14.2 was used to pool prevalence estimates and effect sizes (odds ratio) on the association between disclosure and adherence to ART. The random-effects model using the method of DerSimonian and Laird was used with the estimate of heterogeneity being taken from the inverse-variance fixed-effect model [39,40]. For the meta-analysis of prevalence, a procedure in STATA, called '*metaprop*', that performs the Freeman-Tukey double arcsine transformation, was used to compute the weighted pooled estimate and perform back-transformation on the pooled estimate [41]. The Higgins $I^2$ statistic was used to describe the percentage of total variability in study estimates that was due to heterogeneity [42]. The $I^2$ statistic values of 25%, 50%, and 75% would mean low, medium, and high heterogeneity, respectively [42]. Subgroup analysis was conducted by disclosure status to present the pooled prevalence of adherence, type of adherence report, and whether primary studies reported an adjusted estimate or not. For the prevalence meta-analysis only cross-sectional studies reporting proportion of children and adolescents adherent to ART medication or knew their HIV status were considered. For the meta-analysis of association between HIV status disclosure and adherence to ART medication, all studies reporting odds ratio or allowed computing from the data were considered. Sensitivity analysis was also conducted to check a single study influence on the overall pooled estimate. Publication bias, which represents the tendency to report positive findings [43], was checked visually by inspecting funnel plot and also objectively by using the Egger's regression test to statistically assess the asymmetry of the funnel plot [44]. In the absence of publication bias, the effect size estimates would distribute evenly around the pooled effect size with a greater variability for small studies [43]. With a non-significant Egger's regression test, there was no publication bias. All statistical tests were declared significant at P-values<0.05.

## Results

### Search results

A total of 1,578 studies were located from database search and three studies were included by hand search. After removing duplicates and excluding irrelevant titles in the abstract and title screening, fifty-seven articles were retained for a full-text review. A total of forty-four articles were excluded with reasons from the full-text review leaving thirteen studies [45–57] for the systematic review (for sixteen disclosure was not reported as exposure variable or effect size cannot be computed, in fifteen outcome of interest was not available, six included adult population, three were qualitative studies, for two articles full-text was not available, one was a book chapter (review), and one had a high risk of bias) (S4 Table). Out of the thirteen studies, one did not provide odds ratio (OR)/or relative risk (RR) (or unable to calculate from the data provided) as a measure of association [51]. Consequently, only twelve studies [45–50,52–57] were included in the meta-analysis (Fig 1). One study, Nabukeera-Barungi et al. [52], reported

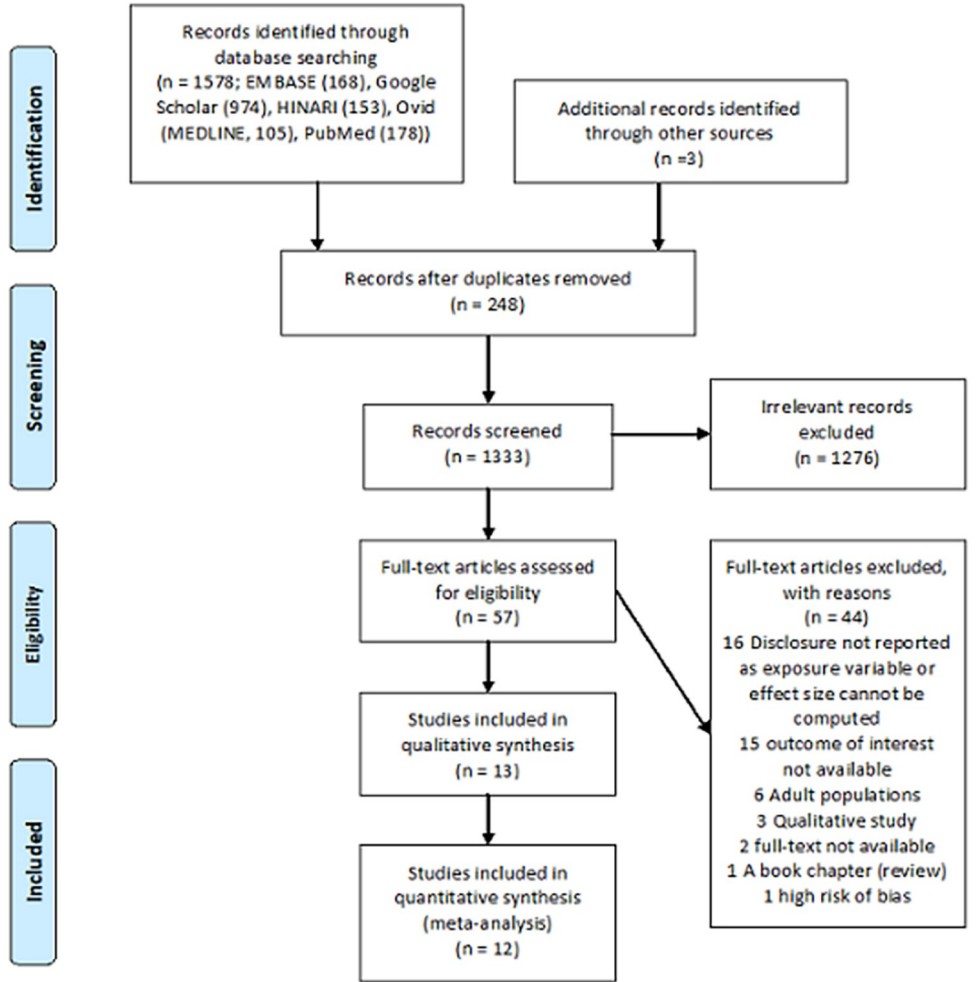

**Fig 1. PRISMA flow diagram showing search results and study selection.**

separate ORs for the effect of child's knowledge of his/her HIV status and knowledge of child's HIV status by other family members in addition to the primary caregiver resulting in 13 separate samples considered for pooling.

A brief summary characteristic of the included studies is presented in Table 1. This meta-analysis pooled ORs from 12 studies on the association between disclosure of HIV status and adherence to ART among children and adolescents living with HIV. Three of the studies were not published in a peer reviewed journals [49,50,54] and the remaining nine were published between 2007 and 2022 [45–48,52,53,55,57]. The primary studies included altogether contributed a total of 4422 subjects with a minimum sample of 98 and a maximum of 873 subjects.

The studies were conducted in five countries in Sub-Saharan Africa, namely: five studies from Ethiopia [45,46,48,49,55], two studies from South Africa [47,56], two studies from Uganda [52,57], one study from Kenya [50], one study from Namibia [54], and one multi-country (Burundi, Democratic Republic of Congo, and Cameroon) [53]. In terms of design nine studies were cross-sectional [45–50,52,54,57], one case-control [55], and two cohort [53,56]. Seven of the studies provided adjusted effect estimates [45,47–49,53,55,56], and others reported (or the authors of this meta-analysis calculated) crude estimates [46,50,52,54,57]. In studies that reported adjusted estimates [45,47–49,53,55,56], the factors controlled for include

**Table 1. Summary of characteristics of the included studies and population studied.**

| Author, year | Country | Study setting and conduct | Inclusion and exclusion criteria | Sample and data collection | Definition of adherence | Children's characterises | Adherence by disclosure status | Effect size and variables adjusted in the model | Risk of Bias |
|---|---|---|---|---|---|---|---|---|---|
| Arage, 2014 [45] | Ethiopia | Across-sectional study was conducted in three hospitals in South Wollo. | Children who received ART for at least one month and aged 2 months to 14 years. | Data were collected in April 2013 from 440 caregivers in a: face-to-face interview | Adherence was based on caregivers' self-report and measured as: ≥95% of medication doses were taken (past one month, past three and seven days. | Children's mean age was 9.4 years (54.8% were in the age range of 10–14 years). | The past one month, seven and three days adherence was 78.6%, 89.8% and 95.9%, respectively. Level of disclosure was 55%, and adherence among the disclosed was 88.5% and 63.3% among the non-disclosed. | Adjusted odds ratio (AOR) = 3.47 (95% CI: 2.10 to 6.81). Effect estimate was adjusted for variables in the multivariable logistic regression model including: caregivers' socio-demographic variables, substance use, knowledge about and attitude towards ART. Medication burden, distance, cd4 count, and care support. | Moderate risk of bias |
| Biressaw, 2013 [46] | Ethiopia | A cross-sectional study hospital-based study was conducted in Addis Ababa | Children aged (<15 years) and received ART for at least three months were included and those in the orphanage, aged less than 3-months, and who were outside of Addis Ababa were excluded. | Data were collected from 210 caregivers through a face-to-face interview. The study was conducted between December 11, 2011 to January 30, 2012. | Adherence data source was unannounced home-based pill count, and adherence was defined as taking ≥95% of doses in the past seven days. | Median age: 11 years (74.8% were 9 years or older). | 42.3% of children did know that they had HIV Adherence was 29.2% among disclosed and 39.7% among the non-disclosed. | Crude odds ratio (COR) = 0.63 (95% CI: 0.35 to 1.13) | Moderate risk of bias |
| Cluver 2015 [47] | South Africa | A community traced multi-centre health facility study was conducted in the Eastern Cape. | Included in the study were adolescents aged 10–19 years, were taking ART, and had a record of their names and addresses. | Data were collected through an interview with 684: adolescents. | Adolescents self-reported on their level of adherence which was defined as taking ≥95% of doses in the past seven days preceding the interview. | The mean age of adolescents in the stud was: 13.4 (52% female and 79% perinatally infected). With regard to disclosure of HIV status, 73.2% of girls and 67.6% of boys knew their HIV status. | The past-week adherence was 64%. The proportion of adolescents who knew their HIV status was 70.2%. | AOR = 2.18 (95% CI: 1.47 to 3.24). The effect size was adjusted for variables including relationship, orphan hood, perinatal infection, medication burden, changes in medication, clinic type, travel time to clinic and lone/accompanied clinic attendance. | Low risk of bias |

*(Continued)*

**Table 1.** (Continued)

| Author, year | Country | Study setting and conduct | Inclusion and exclusion criteria | Sample and data collection | Definition of adherence | Children's characterises | Adherence by disclosure status | Effect size and variables adjusted in the model | Risk of Bias |
|---|---|---|---|---|---|---|---|---|---|
| Dachew 2014 [48] | Ethiopia | A Health facility-based (one hospital and a poly clinic) cross-sectional study was conducted in Gondar, Northwest Ethiopia. | Children aged 2 months to 15 years and received HAART for at least 2 months. | Data were collected by a face-to-face interview with 314 caregivers from January to March 2012. | Adherence measurement was based on caregivers' self-report. The adherence definition used was taking ≥95% of medications in specified period of time. | children whose adherence data collected were 56.1% were boys and 54.7% were between the ages of 10–15 years while 8.9% were below 4 years of age. For 66.9% of the children, the primary caregivers were biological parents. | The Past 3, 7, and 30 day adherence was 98.7%, 96.8%, and 90.4%, respectively. A total of 50.3% of children knew their HIV status and adherence was 86.8% among disclosed and 95.4% among the non-disclosed. | AOR = 0.27 (95% CI: 0.24 to 0.32). The effect estimate was adjusted for variables including child age, caregiver's education, employment status and knowledge of ART medication. | Moderate risk |
| Nabukeera-Barungi 2007 [52] | Uganda | A cross-sectional study was conducted in Mulago Hospital paediatric HIV clinic, Uganda. | Included were = children aged 2–18 years, within 20-kilometers radius, took ART for at least 1 month. The excluded were siblings had already been enrolled, coming to clinic without pill boxes, and used ARV syrup formulation. | Interviews were conducted with 170 children and their caregivers. Children over 12 years self-reported their adherence status. | Adherence was based on self-reported adherence (≥95%) of past three and seven days and clinic-based pill count (four weeks) and unannounced home-based pill count (2–3 weeks after enrolment). Home-based unannounced pill count served as a standard. | Females (57.1%), 51.8% were in the age range of 10–18 years and 10.8% were under-five years of age with a median age of 10 years. 82.4% of children had been on ART for one year. | Based on the ≥95% adherence definition: adherence was 94.1% based on clinic-based pill count, 89% on past three days report, and 72% based on unannounced home-based pill count. 59.7% of children knew their HIV status. | cOR = 1.10 (95% CI: 0.48 to 2.51), for the sample of 114 eligible children (8 years and above). When someone else besides the caregiver knows status (n = 164), cOR = 3.34 (95% CI: 1.13 to 9.82). | Low risk of bias |
| Tjituka, 2018 [54] | Namibia | A cross-sectional study was conducted in Katutura State Hospital in Namibia. | Adolescents aged 10–19 years and taking ART for at least twelve months were enrolled into the study based on convenience sampling. | Data were collected using a face-to-face interview with 200 adolescents during their clinic visit. | Adolescent self-reported adherence status (≥ 95% of the prescribed ARV drugs) on three consecutive visits. | Of the 200 study participants, 59.1% were females and the mean age of the participants was 12.1 years. | 97.5% satisfied the adherence cut-off. Overall, 49.5% of the total participants were aware of their HIV status. | cOR = 1.48 (95% CI: 0.24 to 9.08). | Moderate risk of bias |
| Kimanthi, 2016 [50] | Kenya | A cross-sectional study was conducted in Kangudo district Hospital. | Adolescents aged 10–19 years and who had been taking ART for at least six months. | Sample size: 98. Data were collected using a face-to-face interview with caregivers. | Adherence was defined as taking ≥95% of prescribed drugs; measure based on self-report. | Participants: 47% were females and 64% were in early adolescent, 10–14 years of age. | Good adherence: 76% based on 3-days report and 55% based 30-days report satisfied. | AOR = 1.08 (0.44 to 2.65); Effect estimate was adjusted for school type, motivation to medication, and influence on daily routines. | Moderate risk of bias |

(*Continued*)

**Table 1.** (Continued)

| Author, year | Country | Study setting and conduct | Inclusion and exclusion criteria | Sample and data collection | Definition of adherence | Children's characterises | Adherence by disclosure status | Effect size and variables adjusted in the model | Risk of Bias |
|---|---|---|---|---|---|---|---|---|---|
| Fikadu, 2013 [49] | Ethiopia | A health-facility based cross-sectional study was conducted in Addis Ababa, Ethiopia. | Adolescents aged 10–18 years and who had been taking ART for at least 12 weeks. | Self-reported data from 396 caregivers were collected in October 2012. | Adherence was defined when ≥95% of prescribed drugs were taken in the past seven days before the survey date. | 45.6% were females and 63% were between 10 to 14 years. | 77.3% had good adherence (based on past 7-days report) and 47% knew their HIV status. | AOR = 9.94 (95% CI: 4.48 to 22.08). Effect estimate was adjusted for adolescent age, sex, and regimen type. | Low risk of bias |
| Newman 2015 [53] | Multi-country (Burundi, Cameroon, and DRC) | Baseline data of a cohort study reported findings relevant to this study. | Adolescents and children aged 5–18 years: 69% of disclosed and 75% of undisclosed children were between the ages of 5–11 years. | Caregiver self-report | Children were non-adherent if missed doses of ART for 2 or more consecutive days in the past 30 days. | Adherence data were available for 165 children at baseline. | 91.4% had never missed ART doses among the disclosed and 88.1% among the undisclosed. Disclosure rate was 50%. | AOR = 0.91 (95% CI: 0.22 to 3.78). The effect estimate was adjusted for country, gender, and age. | Moderate risk of bias |
| Montalto 2017 [51] | Kenya | A retrospective cohort study was conducted at Kericho District Hospital (KDH) | Adolescents aged 9–19 years and received care between 1 April 2004 and 1 November 2012, and had at least one pre-disclosure and post-disclosure visit. | Data were collected through clinical chart review for 96 adolescents. | Caretaker or patient self-report through clinical interviews (Morisky Medication Adherence Scale), pill count, and pharmacy refills were used to define adherence. | 56.3% were girls and were in the age range of 9.21 to 17.05 years, with a mean of 12.34. | Study outcome was patient mean percentage adherence. There was a change in level of adherence from 80.2% in pre-disclosure to 91.7% during post-disclosure | NA | Low risk of bias |
| Mengesha 2022 [55] | Ethiopia | A Multi-centre health-facility based case-control study was conducted in Dire Dawa, Eastern Ethiopia | Perinatally HIV-infected children of age 6–17 years and were on ART for at least 6 months and currently active. | A total of 272 data points (78 cases and 194 controls) constituted the study size. Data were obtained clinical record reviews and face-to-face interview with caregivers. | Cases definition → took <95% of prescribed ART drugs. Control definition → took ≥ 95% of prescribed ART drugs. The Case and control definitions are based on physician's evaluation of adherence. | The cases and controls did not differ by their caregiver socio-demography. The children's median age was 14 years. Females accounted 56.3%, and for 83.1%, the reporting caregivers were biological parents. | Children who knew their HIV status were 61.8% andwhen stratified by case-control status: 72.2% of controls were aware of their HIV status while only 35.9% of cases were aware of their HIV status | aOR = 3.32 (95% CI: 1.4 to 7.92). The Effect estimate was adjusted for caregiver's data including substance use, relation with child, monthly family income, HIV status; and child related data including age, ART dose, ART regimen, and viral load change from baseline. | Low risk of bias |

*(Continued)*

**Table 1.** (Continued)

| Author, year | Country | Study setting and conduct | Inclusion and exclusion criteria | Sample and data collection | Definition of adherence | Children's characterises | Adherence by disclosure status | Effect size and variables adjusted in the model | Risk of Bias |
|---|---|---|---|---|---|---|---|---|---|
| Edun 2022 [56] | South Africa | A multi-centre health facility and community traced prospective cohort study was conducted in the Eastern Cape | All adolescents aged 10–19 years and initiated ART at age ≤10 years and had a record of their name and addresses were eligible for inclusion. | The Patient Medication Adherence Questionnaire was used for interview with 813 adolescents and also clinical records were reviewed. | Adherence was based on not missing any of the ART medication doses in the past 7 days preceding the survey. | A 1:1 ratio of boys and girls were represented in the study. The median age of eligible participants was 13, 14 and 15 years at rounds 1, 2 and 3, respectively. | Adherence improved among undisclosed between Round 1 & Round2, whereas it decreased among disclosed during the same period | AOR = 1.04 (95% CI: 0.8 to 1.36). The effect estimate was adjusted age, age at ART initiation, sex, dwelling type, caregiver relationship, orphan hood status, anticipated and secondary stigma and study round. | Low risk of bias |
| Kairania 2022 [57] | Uganda | A multi-centre cross-sectional study was conducted in 42 facilities in Masaka region | Included in the study were adolescents aged 12–17 years and had taken ART for at least 6 months prior to the study. | Data were collection in a face-to-face interview with 524 adolescents. Name of the condition was explored if adolescents knew their condition. | Taking HIV medication as directed (right dose, correct time, every time) within the previous 30 days of the interview was deemed to be ART adherence. | 60.6% were in the age range of 12–14 years and girls accounted 51.9% of the study participants.85.9% received disclosure and 45.3% adherent to ART. | Adherence to ART by disclosure of HIV status was 43.7% among disclosed versus38% among undisclosed. | cOR = 1.26 (95% CI: 0.75 to 2.12) | Low risk of bias |

age (both caregiver and child), caregiver's sex, country, education and school type, knowledge and attitude towards ART medication, motivation for medication, regimen type, caregiver's substance use, family income, viral load change, child medication burden, distance travelled, and orphan hood.

Adherence was measured either using self-report/clinical record review in nine studies [45,47–50,53,55–57] or using pill count only in one study[54] or using mix of the two methods in two studies [46,52] based on the ≥95% cut-off which represent the proportion of prescribed ARV drugs that were taken during the period of assessment considered. Seven studies [47,49,50,52,54,56,57] measured adherence data from only adolescent population whereas the remaining six studies [45,46,48,52,53,55] measured from a mixed population (adolescents and children below the age of 10 years and as young as 2 months). Except in Dachew et al. [48], that did not provide average age of children studied, the average age of children studied ranged from 9.4 years to 13.4 years. A retrospective study by Montato et al. [51] reported patient mean ART adherence percentage based on number of clinic visits for a patient to be deemed adherent and the outcome was measured as a continuous variable. The study reported a higher proportion of adherents during post-disclosure compared to the pre-disclosure period (Table 1).

## Proportion of adherence by disclosure status

The overall pooled prevalence of HIV status disclosure was 58% and the corresponding overall pooled prevalence of good adherence was 73%. When stratified by HIV disclosure status, the

**Table 2. Pooled prevalence of adherence to anti-retroviral therapy by HIV disclosure status.**

| Characteristic estimated | Number of studies | Pooled prevalence, % (95% CI) | I² (%) | P-value |
|---|---|---|---|---|
| Proportion aware of HIV status | 8 | 58.0 (46.0 to 70.0) | 97.2 | <0.001 |
| Overall proportion of good ART adherence | 8 | 73.0 (56.0 to 87.0) | 98.63 | <0.001 |
| Proportion of good adherence among disclosed | 8 | 77.0 (56.0 to 92.0) | 98.34 | <0.001 |
| Proportion of good adherence among non-disclosed | 8 | 69.0 (51.0 to 85.0) | 96.75 | <0.001 |
| Adherence among adolescent only who aware of their HIV status | 5 | 80.0(51.0 to 98.0) | 98.59 | <0.001 |

ART = antiretroviral therapy; CI = confidence interval; HIV = human immunodeficiency virus.

proportion of adherents to ART among those who knew their HIV status was 77% versus 69% among those who did not know (Table 2).

## Association between HIV diagnosis disclosures and adherence to ART

Meta-analysis of results from thirteen independent samples indicated a pooled effect estimate of 1.60 (OR = 1.60, 95% CI: 0. 81 to 3.14; $I^2$ = 95.6%, P = <0.0001). Noting a significant heterogeneity in the pooled estimate, we conducted a sensitivity analysis by removing one study at a time and observed whether there was an improvement on the observed heterogeneity. Consequently, we noted that removing a study by Dachew et al. [48] resulted in a substantial improvement on the observed heterogeneity (S5 Table). This was further assessed graphically using an influence analysis (Fig 2). The overall pooled effect size that we obtained by removing this study was 1.88 (OR = 1.88, 95% CI: 1. 21 to 2.94; $I^2$ = 79.8%, P = <0.0001) (Fig 3).

## Subgroup analysis

Awareness of HIV status had a positive association with adherence to ART in both primary studies that included only adolescents and those with mixed (adolescents and younger

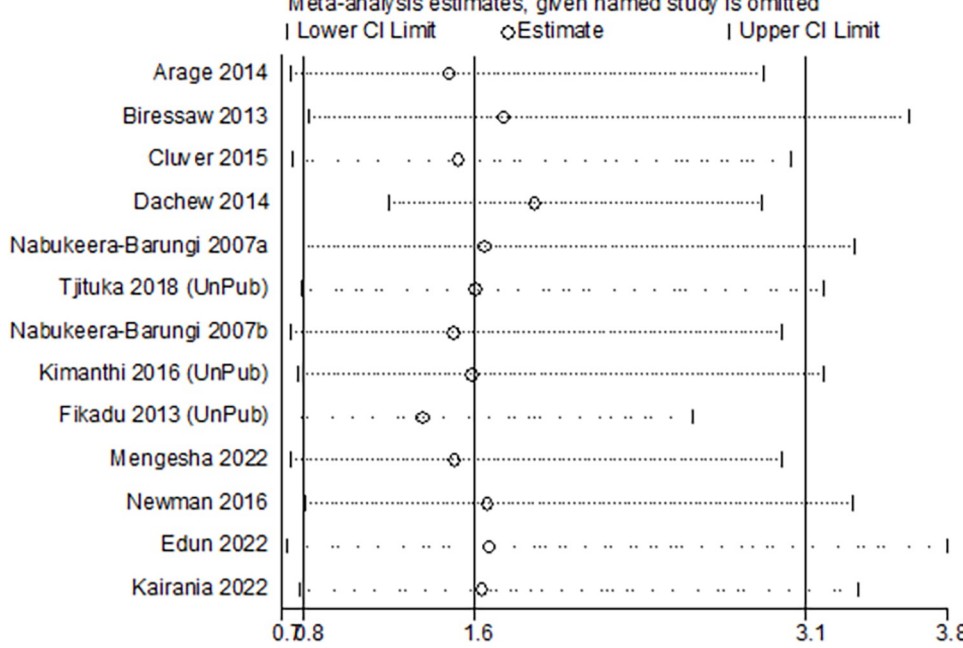

**Fig 2. The influence graph for the meta-analysis of results from 13 independent samples on the association between disclosure of HIV status and adherence to ART.**

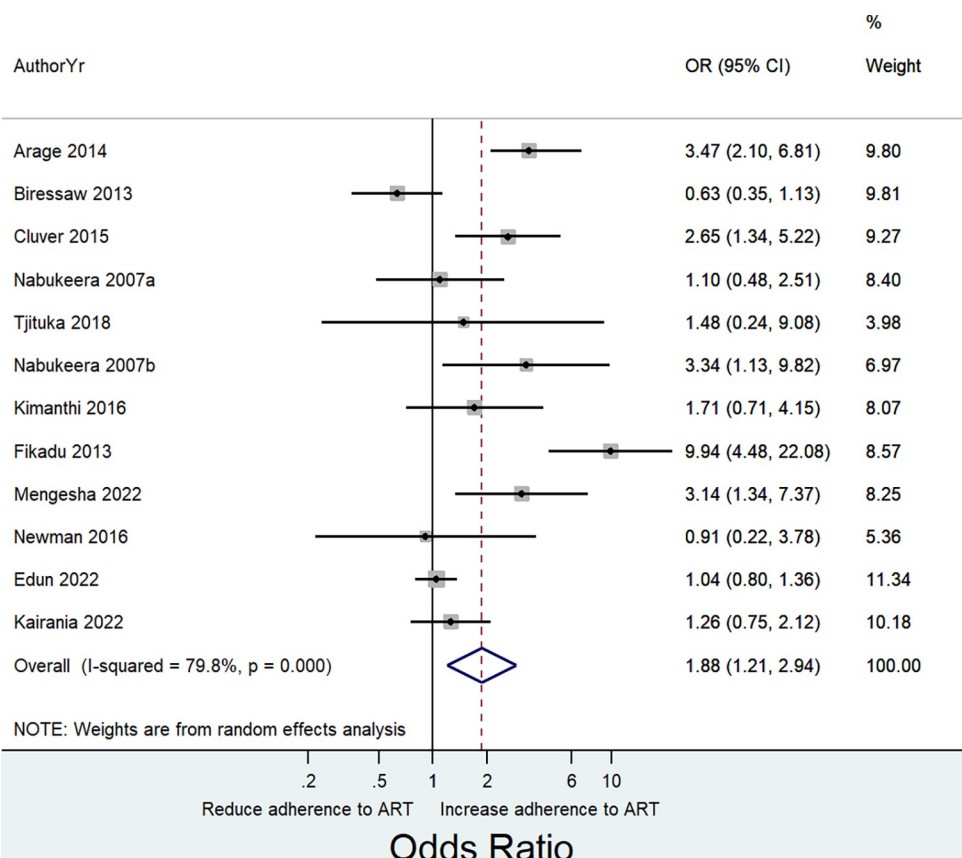

**Fig 3. Forest plot showing the pooled effect size of the association between HIV disclosure status and adherence to ART based on random effects model among HIV infected children and adolescents.**

children) population. However, the pooled effect estimate was significant only among primary studies that included the adolescent only subpopulation: (adolescent only subpopulation, seven studies, Pooled OR = 1.89, 95% CI: 1.06 to 3.37; $I^2$ = 81.3%, P = <0.0001; mixed subpopulation, five studies, pooled OR = 1.86, 95% CI: 0.80 to 4.32; $I^2$ = 79.8%, P = <0.0001 (Fig 4). We further did a subgroup analysis by whether primary studies provided adjusted effect estimates for confounding factors. Accordingly, adherence to ART had a positive and significant association with awareness of HIV status when pooling considered studies that reported adjusted effect estimates (six studies, Pooled OR = 2.61, 95% CI: 1.22 to 5.57; $I^2$ = 88.1%, P = 0.001) (Fig 5).

## Publication bias

There was no evidence of publication bias as presented in the funnel plot (Fig 6), and it was further evidenced by the Egger's test for small-study effects (Coef = -0.140, 95% CI: -0.999 to 0.719; P = 0.724; number of studies = 12).

## Discussion

A global meta-analysis reported 84% level of adherence to ART among adolescents of the African and Asian descents [58]. The report for the African and Asian descent adolescents was similar to the level of adherence reported in this meta-analysis for the adolescent only

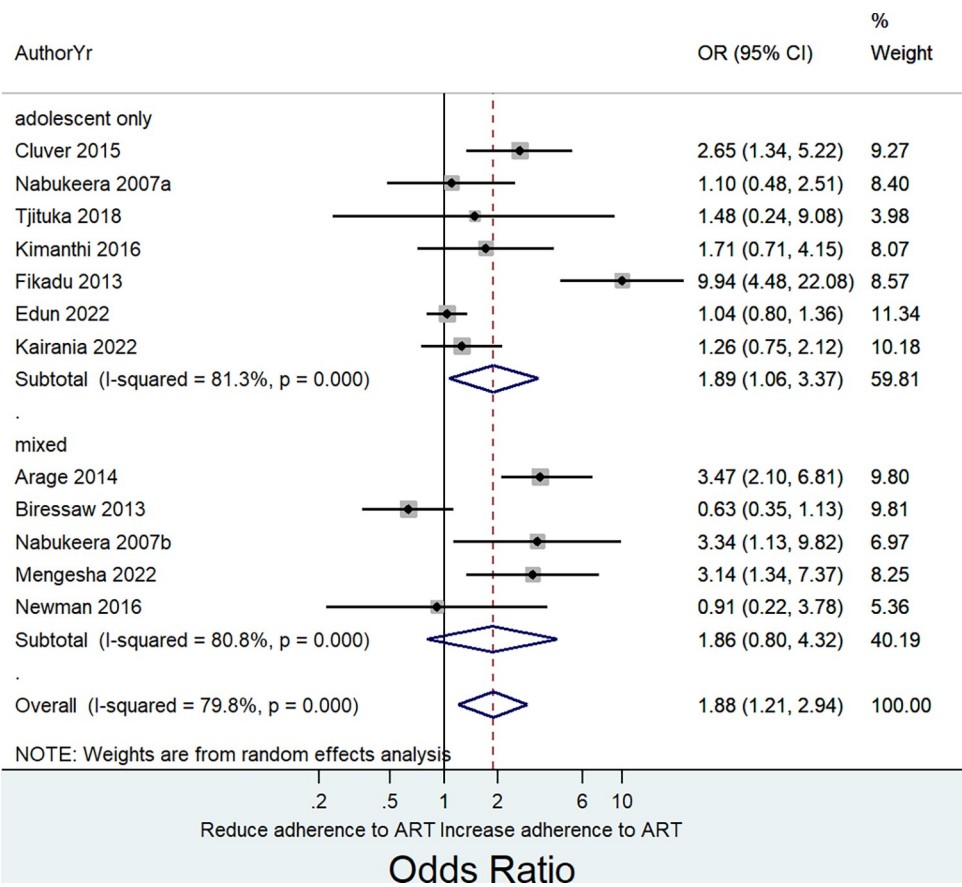

**Fig 4. Subgroup analysis by age category (adolescent only subpopulation versus both children and adolescent subpopulation) showing the pooled effect size of the association between HIV disclosure status and adherence to ART based on random effects model among HIV infected children and adolescents.**

subgroup, 80% (95% CI: 51.0 to 98.0) [58]. However, the level of adherence in the total population of adolescents and younger children, 73%, was lower in the current study. Previous research evidence pointed a regional difference in level of adherence to ART (North America (53%), Europe (62%), and South America (63%)) and also reasoned medication fatigue as a possible factor for the observed regional differences [58,59]. In addition to medication fatigue, caregiver's failure to closely supervise medication time could result in poor adherence as younger children had low autonomy with their medication [60].

The level of good adherence to ART is expected to vary depending on existence of different factors in a complex relationship. A review of factors associated with adherence to ART among adolescents reported a complex web of factors: 44 barriers and 29 facilitators [60]. Knowledge of HIV status was reported as one of the factors that facilitated adherence while stigma, lack of assistance, and forgetfulness were reported as barriers to adherence [60]. Compared to the vital role of good adherence (high levels of compliance to the daily ART dosing ≥95%) for a sustained improvement in survival, viral load suppression, and reduction in onward transmission [12,61], the pooled prevalence of adherence reported in this study signals that a long walk still remains to keep the gains achieved so far.

This meta-analysis nailed that children and adolescents who were aware of their HIV positive status had a positive and significant association with adherence to ART compared to those who were not aware. A systematic review on the disclosure of HIV status to children in

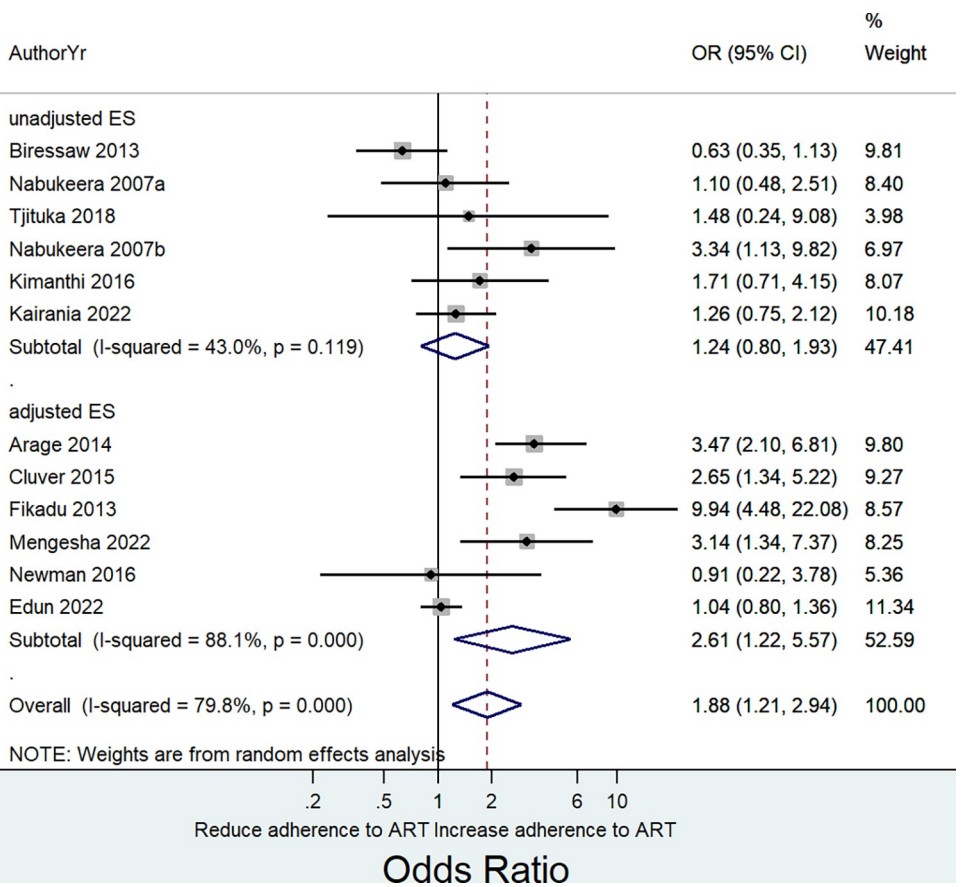

**Fig 5. A subgroup analysis of the association between awareness of HIV status and adherence to ART based on random effects model among HIV infected children and adolescents by whether effect sizes were adjusted.**

resource limited settings reported that adherence to ART was improved during post-disclosure [62] and knowledge of one's HIV status was reported as a facilitator of adherence to ART [60]. Whereas, another systematic review reported that there was no consensus on the effect in any direction (negative or positive) that disclosure had on adherence [29].

The approach disclosure took place might have a moderating effect on adherence to ART. Caregivers often consider disclosure of HIV status to children as a onetime event but not as a process [62]. In addition to the other factors [60], how disclosure took place plays an important role on the physical and psychological wellbeing, which in turn could affect post-disclosure adherence of children and adolescents [62].

Disclosure impacts adherence by creating opportunities to access adherence support and other forms of psychosocial support from family members and peers [63]. Research evidence on impact support groups indicated that it helped to promote adherence by improving adolescents' knowledge and confidence [64]. To reap the beneficial effects of knowledge of one's HIV status on adherence, interventions should address barriers of disclosure at different levels identified in the literature. To mention a few individual level factors, caregiver and child related factors included fear that child may have negative psychological consequences, caregiver's disclosure self-efficacy and deception, fear that child will tell others, lack of social support and forgetfulness, and stigma and discrimination [60,62,65,66]. Therefore, as disclosure could open up avenues for psychosocial support and adherence [63], areas of challenges on its

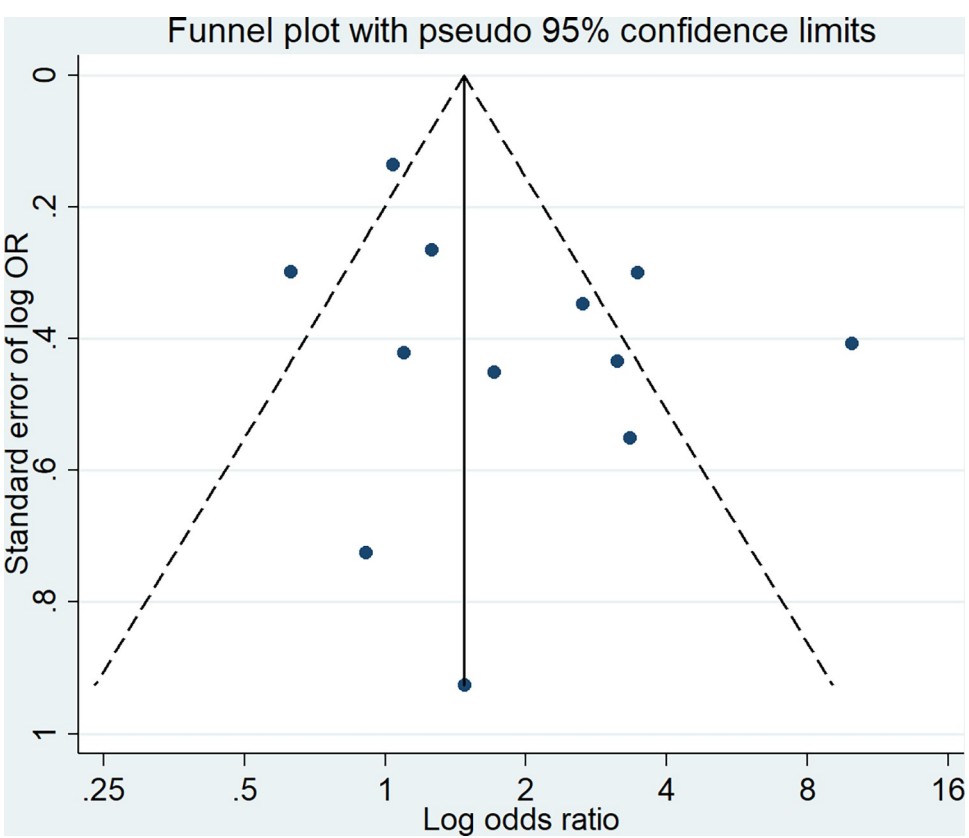

**Fig 6. Funnel plot for a visual assessment of publication bias.**

appropriate implementation as a process should be identified to improve outcomes of the HIV/AIDS care and treatment.

This systematic review and meta-analysis revealed a pooled beneficial effect of knowledge of one's HIV status on adherence to be consistent with previous research evidence [60,62]. Reports that disclosure was associated with physical and emotional difficulties during the immediate post-disclosure period was not conclusive [62]. Furthermore, initial problems with adherence during immediate post-disclosure did go away with a large family support, and as a result, adherence gets improved during post-disclosure [67]. Besides family support, patient's knowledge and self-motivation, among other facilitators, was reported to enhance adherence in qualitative study among adolescents [68].

## Strengths and limitations of the study

The strength of this meta-analysis was that we included both published and unpublished studies through a comprehensive search of major databases, conducted sensitivity and subgroup analysis and reported publication bias based on both visual assessment and also by the egger's regression. Additionally, we employed the Freeman-Tukey Double Arcsine Transformation, which stabilizes the variance, to generate the pooled prevalence estimates. To aid the review process, we used the COVIDENCE software that reduced possible errors and also bias. However, our study was not without limitations as the primary studies could not indicate the temporal sequence of the reported association, we did not conduct a meta-regression to identify sources of heterogeneity due to limited number of studies located, and participants for whom

adherence information was collected were mixed (both children and adolescents) where for the younger children the responsibility of adherence falls on the caregiver and this may mask the true effect of disclosure on adherence.

## Conclusion

This meta-analysis discovered a beneficial correlation between adherence to ART treatment and knowledge of HIV status, particularly among adolescents. When compared to children and adolescents who did not know their HIV status, those who knew their status had a greater pooled percentage of people who took their ART as prescribed. Therefore, raising healthcare professionals' capacity for tailored disclosure assistance and raising caregiver's disclosure self-efficacy could help children's commitment to consistently take their ART medication.

## Supporting information

**S1 Table. PRISMA 2020 main checklist.**
(DOCX)

**S2 Table. Search strings used for a comprehensive search in major databases.**
(DOCX)

**S3 Table. Study quality assessment results for case-control, cohort and cross-sectional studies.**
(DOCX)

**S4 Table. Reasons for the excluded primary studies from the review.**
(DOCX)

**S5 Table. Sensitivity analysis of primary studies included in the meta-analysis.**
(DOCX)

**S6 Table. Dataset.**
(DTA)

## Author Contributions

**Conceptualization:** Melkamu Merid Mengesha.

**Data curation:** Melkamu Merid Mengesha, Awugchew Teshome, Dessalegn Ajema, Abera Kenay Tura.

**Formal analysis:** Melkamu Merid Mengesha.

**Investigation:** Melkamu Merid Mengesha, Awugchew Teshome, Dessalegn Ajema, Abera Kenay Tura.

**Methodology:** Melkamu Merid Mengesha, Awugchew Teshome, Dessalegn Ajema, Abera Kenay Tura, Inger Kristensson Hallström, Degu Jerene.

**Project administration:** Melkamu Merid Mengesha.

**Resources:** Melkamu Merid Mengesha, Awugchew Teshome, Dessalegn Ajema, Abera Kenay Tura, Inger Kristensson Hallström, Degu Jerene.

**Software:** Melkamu Merid Mengesha, Awugchew Teshome, Dessalegn Ajema, Abera Kenay Tura, Inger Kristensson Hallström, Degu Jerene.

**Supervision:** Melkamu Merid Mengesha, Inger Kristensson Hallström, Degu Jerene.

**Validation:** Melkamu Merid Mengesha, Awugchew Teshome, Dessalegn Ajema, Degu Jerene.

**Writing – original draft:** Melkamu Merid Mengesha, Awugchew Teshome, Inger Kristensson Hallström, Degu Jerene.

**Writing – review & editing:** Melkamu Merid Mengesha, Awugchew Teshome, Inger Kristensson Hallström, Degu Jerene.

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
