## [Decision Letter · Decision Letter 0]

21 Oct 2022

PONE-D-21-36858The association between HIV Diagnosis Disclosure and Adherence to Anti-retroviral Therapy among Adolescents Living with HIV in Sub-Saharan Africa: A Systematic Review and Meta-AnalysisPLOS ONE

Dear Melkamu Merid Mengesha,

Thank you for submitting your manuscript to PLOS ONE. After careful consideration, we feel that it has merit but does not fully meet PLOS ONE’s publication criteria as it currently stands. Therefore, we invite you to submit a revised version of the manuscript that addresses the points raised by the reviewers below.

Please submit your revised manuscript by Dec 01 2022 11:59PM If you will need more time than this to complete your revisions, please reply to this message or contact the journal office at plosone@plos.org. Please include the following items when submitting your revised manuscript:

We look forward to receiving your revised manuscript.

Kind regards,

Sharada Prasad Wasti, MHCM,  MSc, Ph.D.

Academic Editor

PLOS ONE

1, Please ensure that your manuscript meets PLOS ONE's style requirements, including those for file naming. The PLOS ONE style templates can be found at

**2. **We noticed you have some minor occurrence of overlapping text with the following previous publication, which needs to be addressed:

- https://link.springer.com/article/10.1186/s12887-018-1330-5

In your revision ensure you cite all your sources (including your own works), and quote or rephrase any duplicated text outside the methods section. Further consideration is dependent on these concerns being addressed.

 Reviewers' comments:

Reviewer's Responses to Questions

**Comments to the Author**

1. Is the manuscript technically sound, and do the data support the conclusions?

Reviewer #1: Partly

Reviewer #2: No

2. Has the statistical analysis been performed appropriately and rigorously? 

Reviewer #1: Yes

Reviewer #2: Yes

3. Have the authors made all data underlying the findings in their manuscript fully available?

Reviewer #1: Yes

Reviewer #2: Yes

4. Is the manuscript presented in an intelligible fashion and written in standard English?

Reviewer #1: Yes

Reviewer #2: Yes

5. Review Comments to the Author

Reviewer #1: ABSTRACT

1. On Line 36, it is unclear who “higher among the disclosed” refers to. Are the disclosed those who have disclosed their status to others, or to whom their own status has been disclosed.

2. I would recommend avoiding causal language like “disclosure of HIV status INCREASED the odds of adherence…”. I would recommend associational language such as “disclosure of HIV status was ASSOCIATED with a higher odds of adherence…” Similarly, “improved adherence” should be avoided unless disclosure was evaluated in some kind of “intervention” where causal inference can be made.

INTRODUCTION

3. On line 86, “Several researches in…” is not clear. Is intended to be “researchers”, or “research studies”?

METHODS

4. The inclusion of both child and adolescent studies seems questionable. The authors did conduct stratified analyses, but the interpretation of the child studies is very different from the adolescent studies.

5. While I don’t have direct evidence, I am very surprised about the small number of studies that were identified in this systematic search. I am concerned that the search terms may have missed relevant article related to status disclosure.

RESULTS

6. Line 288-291: Test for publication bias have low power. Therefore, it is inappropriate to say that there was no publication bias. Rather, you can only say that you found no evidence of publication bias.

DISCUSSION

7. The discussion is quite general and difficult to follow in places. The discussion should focus on interpreting the findings and placing them in context, rather than restating want was already presented in the results section.

Reviewer #2: This paper describes a systematic review and meta-analysis of HIV status knowledge and adherence among adolescents/children in sub-Saharan Africa. It is an important topic and is methodologically sound, with no flaws in the conduct of the review and analysis.

The main concerns are the small sample of only 10 or so eligible studies. There were qualitative and quantitative studies but the results were not reported separately for the two groups. The use of the terms “HIV status disclosure” is a bit misleading. It’s really awareness of knowledge of one’s own status that is the focus of this article. The differences between children and adolescence in terms of disclosure to the patient of the HIV status and adherence behaviors are huge. Frankly, it is hard to reconcile analyses that include both groups.

The biggest issue has to do with the assumption of causality. Only in one section do the authors note there was no way to control for the timing of the two variables of adherence and knowledge of status. It is very likely that parents and providers were very selective in whom they chose to reveal a diagnosis. This is a major point and one that negates that blanket recommendation that efforts should be made to enhance disclosure to youth. Also, several studies have pointed to difficulties youth experience when first informed of their status; these issues should not be undercounted.

6. PLOS authors have the option to publish the peer review history of their article (what does this mean?). If published, this will include your full peer review and any attached files.

Reviewer #1: No

Reviewer #2: **Yes: **Jane M. Simoni

---

## [Author Response · Author response to Decision Letter 0]

5 Dec 2022

Manuscript number: PONE-D-21-36858

Title: The association between HIV Diagnosis Disclosure and Adherence to Anti-retroviral Therapy among Adolescents Living with HIV in Sub-Saharan Africa: A Systematic Review and Meta-Analysis

Dear Editor,

On behalf of all the authors of the manuscript entitled above, I want to extend my sincere thanks to the editor and reviewers for the invaluable time and expertise invested in our manuscript which helped us improve our manuscript. We considered all the comments and suggestions and provided a point-by-point response to reviewers’ questions and concerns. We indicated the changes made in track changes and also uploaded both the clean and the copy with track-changes in the revised submission. Following the comments from both the editor and reviewers, we managed to improve the manuscript from the original submission and also updated the search to include additional studies. 

Reviewer #1: ABSTRACT

1. On Line 36, it is unclear who “higher among the disclosed” refers to. Are the disclosed those who have disclosed their status to others, or to whom their own status has been disclosed.

Thank you for the comment on the clarification. The statement, “Higher among the disclosed” was to refer to those who knew their HIV status. The text was now edited in the revised submission as, “…it was higher among adolescents who were aware of their HIV status.”

2. I would recommend avoiding causal language like “disclosure of HIV status INCREASED the odds of adherence…”. I would recommend associational language such as “disclosure of HIV status was ASSOCIATED with a higher odd of adherence…” Similarly, “improved adherence” should be avoided unless disclosure was evaluated in some kind of “intervention” where causal inference can be made.

Thank you for the important suggestion to avoid terms implying causality. We acknowledged the comments and corrected the suggestions accordingly. 

INTRODUCTION

3. On line 86, “Several research in…” is not clear. Is intended to be “researchers”, or “research studies”?

Thank you for the comment. It was to refer to “several research studies in…” we corrected the edit in the revised submission as, “Several research studies in resource-limited settings have assessed the association between disclosure status and levels of ART adherence.”

METHODS

4. The inclusion of both child and adolescent studies seems questionable. The authors did conduct stratified analyses, but the interpretation of the child studies is very different from the adolescent studies.

Thank you for the comment; we acknowledge the comments and the limitation in our study in this regard. In this analysis, we included primary studies that reported association between disclosure of children’s HIV status and adherence to ART. The WHO recommends a developmentally appropriate disclosure of HIV status to children starting at age of 6 years and full disclosure at age 12 years (Available at: https://www.who.int/hiv/pub/hiv_disclosure/en/). Therefore, our study is significantly limited to children aged 6 years and above and adolescents. We excluded studies whose population group is mostly below the age of 6 years. Regarding adherence, both children and adolescents are mostly under supervision of their caregivers, but the extent of supervision could be more flexible among adolescents. Where the primary authors reported separate analysis for children between 6-10 years and adolescents (10-19 years), we extracted effect sizes separately. However, when the estimate came from a mixed group from the outset, it was difficult to separate, and hence we mentioned this submission as a limitation in the revised.

5. While I don’t have direct evidence, I am very surprised about the small number of studies that were identified in this systematic search. I am concerned that the search terms may have missed relevant article related to status disclosure.

Thank you for the comment. We conducted a comprehensive search in major databases including EMBASE, PubMed, ovid/MEDLINE, HINARI, and Google Scholar. Supplementary Table 2 presents the search strings used in each of the databases. Following the comment, we updated our search (from December 31, 2021) and added two new articles published since January 2022 (Last date of hit: November 12, 2022).

Database

PubMed

Kairania R, Onyango-Ouma W, Ondicho TG, Kigozi G. HIV status disclosure and antiretroviral therapy adherence among children in Masaka region, Uganda. Afr J AIDS Res. 2022 Oct;21(3):251-260. doi: https://doi.org/10.2989/16085906.2022.2060843. Epub 2022 Sep 15. PMID: 36111384.

Database

HINARI

Edun, O., Shenderovich, Y., Zhou, S., Toska, E., Okell, L., Eaton, J.W. and Cluver, L. (2022), Predictors and consequences of HIV status disclosure to adolescents living with HIV in Eastern Cape, South Africa: a prospective cohort study. J Int AIDS Soc., 25: e25910. https://doi.org/10.1002/jia2.25910

The authors do agree the limitation on the number of studies identified as higher number of studies increase the power of the study. However, given limited number of primary studies that observed association between HIV status disclosure and its effect on adherence to ART among children (including adolescents), the number of studies is not too small to produce a reliable estimate as we included 12 primary studies in the meta-analysis. To support our statement, we provided a work by Cheung et al. (doi: https://doi.org/10.1007/s11065-016-9319-z). In their publication, among other things, Cheung et al. discussed the number of primary studies needed to conduct a meta-analysis, and different factors affect decision on the number of studies required including discipline specific context, the model used (fixed- or random-effects model), and other considerations. They presented different studies that reviewed several published meta-analyses and reported the number of studies varies across published meta-analysis, from a median of 3 to a median of 35. 

RESULTS

6. Line 288-291: Test for publication bias have low power. Therefore, it is inappropriate to say that there was no publication bias. Rather, you can only say that you found no evidence of publication bias.

Thank you for the comment. We corrected the statement as suggested in the revised submission:” There was no evidence of publication bias as presented in the funnel plot (Figure 6), and it was further evidenced by the Egger’s test for small-study effects (Coef=-0.05, 95% CI: -1.03, 0.94; P-value=0.921; number of studies=12).”

DISCUSSION

7. The discussion is quite general and difficult to follow in places. The discussion should focus on interpreting the findings and placing them in context, rather than restating want was already presented in the results section.

Thank you for the comment. We have now carefully relooked the discussion and revised the interpretation as suggested.

Reviewer #2: This paper describes a systematic review and meta-analysis of HIV status knowledge and adherence among adolescents/children in sub-Saharan Africa. It is an important topic and is methodologically sound, with no flaws in the conduct of the review and analysis.

The main concerns are the small sample of only 10 or so eligible studies. There were qualitative and quantitative studies, but the results were not reported separately for the two groups. The use of the terms “HIV status disclosure” is a bit misleading. It’s really awareness of knowledge of one’s own status that is the focus of this article. The differences between children and adolescence in terms of disclosure to the patient of the HIV status and adherence behaviors are huge. Frankly, it is hard to reconcile analyses that include both groups.

Thank you for the comment. The text referring “qualitative syntheses” in the document is to mean a simple description of summaries from primary studies without a statistical treatment. Regarding the use of the term ‘disclosure’, we replaced it by equivalent words as suggested like “awareness of ones HIV status or knowledge of ones HIV status.” Finally, regarding the use of mixed population, we have tried to conduct a subgroup analysis by age presented findings accordingly. 

The biggest issue has to do with the assumption of causality. Only in one section do the authors note there was no way to control for the timing of the two variables of adherence and knowledge of status. It is very likely that parents and providers were very selective in whom they chose to reveal a diagnosis. This is a major point and one that negates that blanket recommendation that efforts should be made to enhance disclosure to youth. Also, several studies have pointed to difficulties youth experience when first informed of their status; these issues should not be undercounted.

Thank you for the important comment. In the revised submission, we proofread the document and carefully tried to avoid terms or statements that otherwise implicate a causal interpretation. 

Regarding diagnosis disclosure, the main objective of this study was to see if children’s and adolescent’s knowledge of their own HIV positive status is associated with adherence to ART medication. Therefore, in this study, our main intention was to study awareness of one’s positive status not self-disclosure of HIV status to others. 

Owning to inappropriate consideration of disclosure as a onetime event, transient physical and psychological problems could occur. However, it is not lasting long as different research evidence have pointed out. In the revised submission, we discussed our findings by considering this situation that happens during the immediate post-disclosure period.

With kind regards,

---

## [Editor Report · Decision Letter 1]

19 Jan 2023

PONE-D-21-36858R1The association between HIV Diagnosis Disclosure and Adherence to Anti-retroviral Therapy among Adolescents Living with HIV in Sub-Saharan Africa: A Systematic Review and Meta-AnalysisPLOS ONE

Dear Dr. Melkamu,

Thank you for submitting your manuscript to PLOS ONE. After careful consideration, we feel that it has merit but does not fully meet PLOS ONE’s publication criteria as it currently stands. Therefore, we invite you to submit a revised version of the manuscript that addresses the points raised during the review process.

**Thank you for submitting the revised version with addressing the reviewer's comments on your **interesting manuscript. Thank you for your efforts in revising and correcting the manuscript where most of the comments were well addressed but few corrections are needed to prior to the publication.

Abstract result section: line 20  capital letter of sub-Saharan Africa – Sub-Saharan Africa and remove the SSA abbreviation from this abstract section. Correct the line 36 CI  and I^2^ presentation including P (value) as per the following structure- 95% CI (confidence interval): 56 to 87; I^2^=98.63%, P = <0.001 and correct accordingly to the entire manuscript.Abstract section key findings should be clearly stated and the last sentence of the result shouldIt is not clear why the PROSPERO  information has been presented at the last of the conclusion section - Systematic review registration: this systematic review and meta-analysis is registered in the International Prospective Register of Systematic Reviews (PROSPERO) and can be accessed online at ( https://www.crd.york.ac.uk/prospero/display_record.php? ID=CRD42020178084 ). So please remove this information from the abstract section.Correct the presentation of lines 105-109 very clearly and concisely which is not clear at a moment.Correct lines 115-116 and write your final included period December 2021 only.Make clear of table 2 ** ** in the exact value not only ** which is not clear at a moment.Correct line 382 keep only the Conclusion and remove the recommendation.Through review and correct the entire reference list and adhere to the referencing consistency referencing as per the Journal guideline and make consistency in the entire references. Please submit your revised manuscript shortly. If you will need more time than this to complete your revisions, please reply to this message or contact the journal office at plosone@plos.org. We look forward to receiving your revised manuscript.

Kind regards,

Sharada P Wasti, Ph.D., MSc

Academic Editor

PLOS ONE

Journal Requirements:

Additional Editor Comments (if provided):

Thank you for submitting the revised version with addressing the reviewer's comments on your interesting manuscript. Thank you for your efforts in revising and correcting the manuscript where most of the comments were well addressed but few corrections are needed to prior to the publication.

• Abstract result section: line 20 capital letter of sub-Saharan Africa – Sub-Saharan Africa and remove the SSA abbreviation from this abstract section.

• Correct the line 36 CI and I2 presentation including P (value) as per the following structure- 95% CI (confidence interval): 56 to 87; I2=98.63%, P = <0.001 and correct accordingly to the entire manuscript.

• Abstract section key findings should be clearly stated and the last sentence of the result should

• It is not clear why the PROSPERO information has been presented at the last of the conclusion section - Systematic review registration: this systematic review and meta-analysis is registered in the International Prospective Register of Systematic Reviews (PROSPERO) and can be accessed online at ( https://www.crd.york.ac.uk/prospero/display_record.php? ID=CRD42020178084 ). So please remove this information from the abstract section.

• Correct the presentation of lines 105-109 very clearly and concisely which is not clear at a moment.

• Correct lines 115-116 and write your final included period December 2021 only.

• Make clear of table 2 ** ** in the exact value not only ** which is not clear at a moment.

• Correct line 382 keep only the Conclusion and remove the recommendation.

• Through review and correct the entire reference list and adhere to the referencing consistency referencing as per the Journal guideline and make consistency in the entire references.
---

## [Author Response · Author response to Decision Letter 1]

15 Feb 2023

Manuscript number: PONE-D-21-36858

Title: The association between HIV Diagnosis Disclosure and Adherence to Anti-retroviral Therapy among Adolescents Living with HIV in Sub-Saharan Africa: A Systematic Review and Meta-Analysis

Dear Editor,

On behalf of all the authors of the manuscript titled above, I want to extend my sincere thanks to the editor for the invaluable comments and suggestions that helped us improve our manuscript. In the revised submission, we corrected all suggested changes and provided a point-by-point response. We indicated the changes made in track changes and also uploaded both the clean and marked copy with track-changes in the revised submission.

Overall, we appreciate the editor's and reviewers' critical suggestions during the review process, which considerably improved our manuscript.

Editor’s comments and suggestions

1. Abstract result section: line 20 capital letter of sub-Saharan Africa – Sub-Saharan Africa and remove the SSA abbreviation from this abstract section.

a. Thank you for the comment. We removed the abbreviation and corrected the capitalization as in “Sub-Saharan Africa”. The changes made can be found in line 20 of the revised submission.

2. Correct the line 36 CI and I2 presentation including P (value) as per the following structure- 95% CI (confidence interval): 56 to 87; I2=98.63%, P = <0.001 and correct accordingly to the entire manuscript.

a. Thank you for the comment. In the revised submission, we corrected presentation of the confidence interval, p-value, and the I2 consistently throughout the document.

3. Abstract section key findings should be clearly stated and the last sentence of the result should

a. Thank you for your input. We revised and reread the abstract to ensure that the findings were presented clearly in a logical manner.

4. It is not clear why the PROSPERO information has been presented at the last of the conclusion section - Systematic review registration: this systematic review and meta-analysis is registered in the International Prospective Register of Systematic Reviews (PROSPERO) and can be accessed online at ( https://www.crd.york.ac.uk/prospero/display_record.php? ID=CRD42020178084 ). So please remove this information from the abstract section.

a. Thank you for the comment. We removed the above text from the methods section and corrected the manuscript.

5. Correct the presentation of lines 105-109 very clearly and concisely which is not clear at a moment.

a. Thank you for the comment. We corrected this in the revised submission and the changes made can be found in lines 100-105.

6. Correct lines 115-116 and write your final included period December 2021 only.

a. Thank you for the comment. We corrected the entry with the last updated search and this can be found in lines 110-111.

7. Make clear of table 2 ** ** in the exact value not only ** which is not clear at a moment.

a. Thank you for the comments. In Table 2, owning to limited number of studies (three studies) in the subgroup analysis for the mixed population (adolescents and younger children) who were aware of their HIV status, the I2 and the corresponding P-value did not show up. In the revised submission, we removed this row in Table 2 with an incomplete estimate as it did not affect interpretation of findings.

8. Correct line 382 keep only the Conclusion and remove the recommendation.

a. Thank you for the comment. Suggestion is acknowledged and corrected. The changes made can be found at line 376.

9. Through review and correct the entire reference list and adhere to the referencing consistency referencing as per the Journal guideline and make consistency in the entire references. 

a. Thank you for the comments. We proofread the manuscript and corrected reference styles and checked that all are cited in the document.

---

## [Editor Report · Decision Letter 2]

14 Mar 2023

PONE-D-21-36858R2The association between HIV Diagnosis Disclosure and Adherence to Anti-retroviral Therapy among Adolescents Living with HIV in Sub-Saharan Africa: A Systematic Review and Meta-AnalysisPLOS ONE

Dear Dr. Mengesha,

Thank you for submitting your manuscript to PLOS ONE. After careful consideration, we feel that it has merit but does not fully meet PLOS ONE’s publication criteria as it currently stands. Therefore, we invite you to submit a revised version of the manuscript that addresses the points raised during the review process.

We look forward to receiving your revised manuscript.

Kind regards,

Sharada P Wasti, Ph.D., MSc

Academic Editor

PLOS ONE

Journal Requirements:

Additional Editor Comments:

Thank you so much for your review of documents but still paper need to through review and professional English proofread for the final submission.

• Correct line 48 - Keywords - sub-Saharan Africa to Sub-Saharan Africa and make consistency in the entire report which is not consistency i.e. para 51, 146, 418,

• Correct Do not repeat the abbreviation multiple times i.e. para 56 – ART, para 61 & 68 - ALHIV

• Make consistency on writing in para 206 to 208 in word rather than single number i.e. six, three, two and one each.

• Remove para 211 paragraph of - See “Additional file 4” for details of excluded studies.

• Make consistency on writing in para 223 to 227 in word rather than single number i.e. five studies, two studies and one study in each country i.e. Kenya, Namibia and multi-countries.

• Remove = from the Table 1 in all boxes except the C/AoR findings box in the second last column and make consistent of the entire Table 1 boxes.

• Remove the repetition of the para 302 to 313 in the discussion section.

• Remove the para 313 Kim et al reference which you have cited at the end of the sentence i.e. 59. I would suggest you please through review the discussion and make clear, consistence, and coherent of your arguments and do through proofread prior to the final submission.

• Could not understand the para 383 and 384 which are empty now. Make this correct.

• Still all references are not well corrected and made consistent i.e. reference number 1 and 38 insert data insert date and make consistency of the following references- 3, 6, 11, 12, 28, 35, 51, 54, 69 first alphabet should be Capital in these Journals, insert web link or how to get access of 49, 50 and 54 references. Insert journal name in reference number 58 and 61.
---

## [Author Response · Author response to Decision Letter 2]

7 Apr 2023

Manuscript number: PONE-D-21-36858R2

Title: The Association Between HIV Diagnosis Disclosure and Adherence to Anti-retroviral Therapy among Adolescents Living with HIV in Sub-Saharan Africa: A Systematic Review and Meta-Analysis

Dear Editor,

On behalf of all the authors of the manuscript entitled above, I want to extend my sincere thanks to you for the invaluable comments and suggestions that helped us a lot to further improve our manuscript. In the revised submission, we considered all suggested changes and provided a point-by-point response. All changes are highlighted in track changes, and I uploaded both the clean and marked copy of the manuscript.

Finally, we appreciate the editor's and reviewers' critical suggestions during the review process, which considerably have improved our manuscript.

Editor’s comments and suggestions

Author response: thank you so much for the detailed comments that helped us to make the reference list complete and accurate. We have corrected all the suggestions made with regard to including web link to references, inconsistencies with journal name, and consistency of citations and the reference list. While checking citations and the references listed, we found four duplicate entries from the previous submission and corrected these in the revised submission. The duplicates in the previous submission were reference #58, #63, #65, and #66. In the revised submission these references correspond to #56, #12, #60, and #29, respectively. Consequently, the total number of references listed is 68 in the revised submission after removing the duplicates. 

Additional Editor Comments:

Thank you so much for your review of documents but still paper need to through review and professional English proofread for the final submission.

• Correct line 48 - Keywords - sub-Saharan Africa to Sub-Saharan Africa and make consistency in the entire report which is not consistency i.e. para 51, 146, 418,

Author response: Thank you for the comment. The keyword “sub-Saharan Africa” in the previous submission is consistently corrected across the document to “Sub-Saharan Africa”.

• Correct Do not repeat the abbreviation multiple times i.e. para 56 – ART, para 61 & 68 – ALHIV

Author response: Thank you for the comment. Acknowledging the comment, we consistently expanded abbreviations on first use only in this revised submission.

• Make consistency on writing in para 206 to 208 in word rather than single number i.e. six, three, two and one each.

Author response: Thank you for the comment. Acknowledging the comment, in the revised manuscript, we corrected entries in number to text consistently in the results section.

• Remove para 211 paragraph of - See “Additional file 4” for details of excluded studies.

Author response: Thank you so much for the comment. We removed the text ‘see additional file 4’ which was a typological error and not coherent within the paragraph presented.

• Make consistency on writing in para 223 to 227 in word rather than single number i.e. five studies, two studies and one study in each country i.e. Kenya, Namibia and multi-countries.

Author response: Thank you for the comment. As suggested above, in the revised manuscript, we corrected entries in number to text consistently in the results section.

• Remove = from the Table 1 in all boxes except the C/AoR findings box in the second last column and make consistent of the entire Table 1 boxes.

Author response: Thank you so much for the comment. As per the suggestion, we removed the ‘=’ sign in Table and reframed the statements.

• Remove the repetition of the para 302 to 313 in the discussion section.

Author response: Thank you for the comment. Our intention of presenting the statements in the lines 302-3013 was to present the key findings that answered the objectives. However, we acknowledged the editor’s concern that it was repetition of the results. As a result, in the revised submission, we removed the statements in lines 302-313 of the previous submission.

• Remove the para 313 Kim et al reference which you have cited at the end of the sentence i.e. 59. I would suggest you please through review the discussion and make clear, consistence, and coherent of your arguments and do through proofread prior to the final submission.

Author response: Thank you for the comment. We acknowledged the comment and removed double referencing both in text and also in citation list.

• Could not understand the para 383 and 384 which are empty now. Make this correct.

Author response: Thank you for the comment. It was to refer to acknowledgement. As it did not apply in our manuscript, we removed the acknowledgement section in the revised submission.

• Still all references are not well corrected and made consistent i.e. reference number 1 and 38 insert data insert date and make consistency of the following references- 3, 6, 11, 12, 28, 35, 51, 54, 69 first alphabet should be Capital in these Journals, insert web link or how to get access of 49, 50 and 54 references. Insert journal name in reference number 58 and 61.

Author response: Thank you so much for the comment that helped us to correct important mistakes in the previous submission. Acknowledging the comments, we corrected capitalizations as needed, journal name consistency, insertion of dates and links to access web documents. For reference #38, however, we could not get date but provided web link to access checklists that we used to evaluate study quality.

With kind regards,

Melkamu Merid

The Corresponding author

---

## [Editor Report · Decision Letter 3]

27 Apr 2023

The Association Between HIV Diagnosis Disclosure and Adherence to Anti-retroviral Therapy among Adolescents Living with HIV in Sub-Saharan Africa: A Systematic Review and Meta-Analysis

PONE-D-21-36858R3

Dear Melkamu Merid Mengesha,

Thank you so much for your thorough review and address of all the comments and suggestions. We’re pleased to inform you that your manuscript has been judged scientifically suitable for publication and will be formally accepted for publication, you’ll receive a formal acceptance letter and your manuscript will be scheduled for publication.

Kind regards,

Sharada P Wasti, Ph.D., MSc

Academic Editor

PLOS ONE

---

## [Editor Report · Acceptance letter]

4 May 2023

PONE-D-21-36858R3 

The Association between HIV Diagnosis Disclosure and Adherence to Anti-retroviral Therapy among Adolescents Living with HIV in Sub-Saharan Africa: A Systematic Review and Meta-Analysis 

Dear Dr. Mengesha:

I'm pleased to inform you that your manuscript has been deemed suitable for publication in PLOS ONE. Congratulations! Your manuscript is now with our production department. 

Kind regards, 

on behalf of

Dr. Sharada P Wasti 

Academic Editor

PLOS ONE